# Selection and Validation of Reference Genes for Gene Expression in *Bactericera gobica* Loginova under Different Insecticide Stresses

**DOI:** 10.3390/ijms25042434

**Published:** 2024-02-19

**Authors:** Hongshuang Wei, Jingyi Zhang, Mengke Yang, Yao Li, Kun Guo, Haili Qiao, Rong Xu, Sai Liu, Changqing Xu

**Affiliations:** Institute of Medicinal Plant Development, Chinese Academy of Medical Sciences and Peking Union Medical College, Beijing 100193, China; hswei@implad.ac.cn (H.W.); zhongyaoxili@163.com (J.Z.); mengke95@163.com (M.Y.); yaoli12138@163.com (Y.L.); kguo@implad.ac.cn (K.G.); qhl193314@163.com (H.Q.); rxu@implad.ac.cn (R.X.)

**Keywords:** *Bactericera gobica*, insecticide resistance, RT-qPCR normalization, stable reference genes

## Abstract

Insecticide resistance has long been a problem in crop pest control. *Bactericera gobica* is a major pest on the well-known medicinal plants *Lycium barbarum* L. Investigating insecticide resistance mechanisms of *B. gobica* will help to identify pesticide reduction strategies to control the pest. Gene expression normalization by RT-qPCR requires the selection and validation of appropriate reference genes (RGs). Here, 15 candidate RGs were selected from transcriptome data of *B. gobica*. Their expression stability was evaluated with five algorithms (Delta Ct, GeNorm, Normfinder, BestKeeper and RefFinder) for sample types differing in response to five insecticide stresses and in four other experimental conditions. Our results indicated that the RGs *RPL10 + RPS15* for Imidacloprid and Abamectin; *RPL10 + AK* for Thiamethoxam; *RPL32 + RPL10* for λ-cyhalothrin; *RPL10 + RPL8* for Matrine; and *EF2 + RPL32* under different insecticide stresses were the most suitable RGs for RT-qPCR normalization. *EF1α + RPL8*, *EF1α + β-actin*, *β-actin + EF2* and *β-actin + RPS15* were the optimal combination of RGs under odor stimulation, temperature, developmental stages and both sexes, respectively. Overall, *EF2* and *RPL8* were the two most stable RGs in all conditions, while *α-TUB* and *RPL32* were the least stable RGs. The corresponding suitable RGs and one unstable RG were used to normalize a target cytochrome P450 *CYP6a1* gene between adult and nymph stages and under imidacloprid stress. The results of *CYP6a1* expression were consistent with transcriptome data. This study is the first research on the most stable RG selection in *B. gobica* nymphs exposed to different insecticides, which will contribute to further research on insecticide resistance mechanisms in *B. gobica*.

## 1. Introduction

With increasing insecticide resistance comes more frequent pesticide use, the re-emergence of pests and more serious pesticide residue problems in the fields [1,2]. Insecticide resistance is currently a problem plaguing the pest control in Chinese herbal medicines [3]. In order to ensure people’s health, the use of chemical pesticides should be reduced or avoided during the cultivation of Chinese herbal medicines. *Lycium barbarum* L. (Solanaceae, Lycium) is a traditional Chinese medicinal herb with a history of more than 2000 years (Figure 1A) [4,5]. Its dried ripe fruits (also named wolfberry, goji berry, gouqizi) (Figure 1B) are used for medicine and food, and have the effect of anti-aging, improving eyesight and so on [5,6]. However, many pests feed on different parts (leaves, flowers, fruits, etc.) of *L. barbarum* L. during its growth, which seriously affects the quality and yield of wolfberries [7,8]. A major insect pest of *L. barbarum* L. is *Bactericera gobica*, with piercing-sucking mouthparts [9]. The pest absorbs food on the young leaves and tender stems (Figure 1C,D), which depletes the nutrients of *L. barbarum* L. and causes the leaves to become yellow and wilt in large areas, resulting in annual yield losses of 20-50% [9]. At present, the control methods of *B. gobica* mainly rely on chemical pesticides [10]. Excessive application of chemical pesticides has increased the resistance of *B. gobica* [10]. However, the insecticide resistance mechanisms in *B. gobica* are still unclear. Further research on understanding of the molecular mechanisms behind the insecticide resistance of *B. gobica* will facilitate more effective management of wolfberry pests and reduce the frequency of pesticide use in the field.

Verification of gene expression by Real-time quantitative PCR (RT-qPCR) is a prerequisite for its functional resolution [11]. The RT-qPCR has been widely used in molecular biology, including research on insecticide resistance mechanisms, insect–host plant interactions and insect adaptation [12]. RT-qPCR data analysis must include the most stable reference genes (RGs) to normalize target gene expression, to optimize the accuracy of quantitative results [13]. Numerous studies have demonstrated that the most suitable RGs differ in different experimental conditions, and no internal RGs can be stably expressed throughout all conditions [10,14,15]. Even in the same insect species, the most stable RGs had a significant difference under different insecticide stresses. For instance, in *Bradysia odoriphaga*, the most suitable RGs were Elongation factor 1α (*EF1α*), beta actin (*β*-*actin*) and Ribosomal protein L10 (*RPL10*) in response to imidacloprid, chlorfluazuron and phoxim, respectively [16]. In addition, *β-actin* is often used as a stable RG in insects, but is not equally or stably expressed in the two closely related species *Bactrocera dorsalis* and *Zeugodacus* (formerly *Bactrocera*) *cucurbitae* under the same conditions [17,18]. Therefore, screening of stable RGs for each experimental condition is a must to ensure the accuracy of RT-qPCR results.

A large number of studies have been conducted to evaluate and validate reference genes from different insect species, especially crop pests, fruit tree pests and forest pests [14,15,16,17,18]. Based on the identification of stable RGs, the molecular mechanisms on host plant localization, oviposition selection and insecticide resistance of more and more important agricultural pests have been elucidated [1,19,20], which will guide the more effective management of pests in the field. In contrast, there are relatively few studies on the RG screening and target gene functions involved in insecticide resistance and olfactory perception in Chinese herbal pests [11]. To our knowledge, there are no reports identifying the most suitable RGs under different insecticide stresses in *B. gobica*.

To understand the molecular mechanism of resistance to neonicotinoids and pyrethroids, 15 candidate RGs widely reported in Hemiptera insects [21,22,23] were selected for stability verification under different insecticide stresses. These RGs were identified from the transcriptomic data of *B. gobica* adults and nymphs. Then the cycle threshold (Ct) values of these genes were measured by RT-qPCR at multiple developmental stages, for both sexes, for different temperatures, for odor stimulation and under five different insecticides stresses (λ-cyhalothrin, Matrine, Abamectin, Imidacloprid and Thiamethoxam). The expression stabilities of those candidate RGs were evaluated using four different algorithms (Delta Ct, NormFinder, BestKeeper and GeNorm) and a comprehensive analysis online software, RefFinder (http://blooge.cn/RefFinder/). The most stable and optimal combination of RGs were determined by GeNorm and RefFinder analyses. A target gene, cytochrome P450 CYP6a1, involved in insecticide metabolism [1], was used to further verify the most stable or unstable RGs between adults and nymphs and under imidacloprid stress by RT-qPCR analysis. The proposed research provides the most stable reference genes of *B. gobica* across different insecticide treatments for further studies on the mechanisms of insecticide resistance in *B. gobica.*

## 2. Results

### 2.1. Verification of Primer Specificity and RT-qPCR Amplification Efficiencies

Fifteen candidate RGs (*β-actin*, *EF1α*, *EF2*, *Ferritin*, *GAPDH*, *α-TUB*, *β-TUB*, *AK*, *GST*, *RPL8*, *RPL10, RPL32*, *RPS11*, *RPS15* and *RPS20*) and one target gene (CYP6a1) of *B. gobica* were selected for RT-qPCR normalization (Table 1). Sequence details and best BLAST hits for these candidate genes are presented in Appendix A. The specific amplification of each primer pair of candidate RGs (Table 1) was confirmed with RT-qPCR. The specificity of each RG primer was validated by melting curve analysis. As shown in Figure 2, the melting curves of all primer sets exhibited a single amplification peak (Figure 2 and Appendix A). The size of amplicons ranged from 92 to 198 bp. The amplification efficiencies (E) for these genes varied from 90.5% for *α-TUB* to 104.4% for *RPS20*, and the correlation coefficients (*R*^2^) varied from 0.992 (*AK*) to 0.999 (*EF1α, RPS11* and *RPS15*) (Table 1).

### 2.2. Expression Profiles of Candidate Reference Genes in B. gobica

The raw cycle threshold (Ct) values of the 15 candidate RGs for RT-qPCR under several different experimental conditions were collected and are shown in Figure 3 and Appendix A. The Ct values displayed a wide range, from 16.16 (*β-actin*) to 33.33 (*RPL32*), in all samples, and the average Ct ranged from 18.34 ± 0.30 (*β-actin*) to 24.69 ± 0.46 (*α-TUB*), indicating that *β-actin* had the highest expression level, whereas *α-TUB* had the lowest expression level in the samples tested. The mean Ct values of *β-actin* for temperature treatment (17.33 ± 0.00) and for sex (17.34 ± 0.00) with the minimum standard error (SE), indicate that this gene was the most stable of the genes tested under different experimental conditions. The most unstable genes were *α-TUB* (24.69 ± 0.46) and *RPL32* (23.69 ± 0.72) in all samples (Figure 3 and Appendix A). Additionally, the mean Ct values of *RPL32* for developmental stages (31.44 ± 0.49) and odor stimulation (28.10 ± 0.57) were larger than other genes’ mean Ct values (Appendix A). The results also showed that *β-actin* exhibited the smallest variation in expression levels, whereas *α-TUB* and *RPL32* displayed the largest variation among all samples.

### 2.3. Stability of Candidate RGs under Different Insecticide Stresses

Using four algorithms (Delta Ct method [24], GeNorm [25], NormFinder [26] and BestKeeper [27]), the expression stability of fifteen candidate RGs was evaluated. Then a result for comprehensive reference gene ranking was obtained to combine the above four algorithms by the RefFinder method [28]. The optimal number and combination of RGs for RT-qPCR normalization was determined by GeNorm (Pairwise value Vn/(n + 1) < 0.15) and RefFinder data [11,28].

#### 2.3.1. Imidacloprid Treatment

*RPL10* was the most stable gene, based on the results of GeNorm and BestKeeper analyses, while *GST* and *AK* were found to be the most stable genes in the other two algorithms (Delta Ct and NormFinder), respectively (Figure 4 and Appendix A). *β-actin* was found to be the most unstable gene, by all analysis methods used (Figure 4 and Appendix A). GeNorm analysis presented the pairwise value V2/3, which was less than 0.15, indicating that the two RGs were the optimal number for normalization of RT-qPCR data (Figure 5A). The RefFinder results indicated the rank order for gene stability was as follows (from the most to least stable): *RPL10*, *RPS15*, *AK*, *GST*, *GAPDH*, *EF2*, *RPS20*, *EF1α*, *α-TUB*, *RPL8*, *RPS11*, *Ferritin*, *β-TUB*, *RPL32*, *β-actin* (Figure 6). Among them, *RPL10* and *RPS15* were considered as the optimal combination RGs for RT-qPCR normalization under imidacloprid stress in *B. gobica* (Figure 5A and Figure 6 and Table 2).

#### 2.3.2. Thiamethoxam Treatment

The Delta Ct and NormFinder data indicated that *RPL10* was the most stable gene. However, GeNorm and BestKeeper results showed that *GAPDH* and *AK* were the most stable genes, respectively (Figure 4 and Appendix A). All analyses showed the *Ferritin* and *α-TUB* were the least stable genes. The rank order for gene stability determined by RefFinder was as follows (from the most to least stable genes): *RPL10*, *AK*, *RPS11*, *RPS15*, *GAPDH*, *EF1α*, *β-actin*, *β-TUB*, *RPS20*, *RPL32*, *EF2*, *GST*, *RPL8*, *α-TUB*, *Ferritin* (Figure 6). Hence, *RPL10* and *AK* were considered as the optimal combination RGs for RT-qPCR normalization under thiamethoxam treatment in *B. gobica*, as determined from GeNorm results (V2/3 < 0.15) (Figure 5A and Figure 6 and Table 2).

#### 2.3.3. λ-Cyhalothrin Treatment

The results for λ-cyhalothrin treatment indicated that *RPL10* (Delta Ct and GeNorm analyses), *RPL32* (NormFinder analysis) and *EF1α* (BestKeeper analysis) were the most stable genes, while *GAPDH* was the least stable gene (Figure 4 and Appendix A). The genes Geomean of ranking values analyzed by RefFinder were listed as follows (from the most to least stable): 1.19 (*RPL32*), 2.34 (*RPL10*), 3.25 (*EF1α*), 3.83 (*RPS11*), 4.16 (*Ferritin*), 5.09 *(EF2*), 6.9 (*RPL8*), 7.52 (*RPS15*), 9.246 (AK), 9.46 (*β-actin*), 9.82 (*RPS20*), 12.00 (*GST*), 13.00 (*β-TUB*), 14.00 (*α-TUB*), 15.00 (*GAPDH*) (Figure 6 and Appendix A). Combined with the analysis results of RefFinder and GeNorm (V2/3 < 0.15), the optimal combination RGs were *RPL32* and *RPL10* for normalization of RT-qPCR target gene expression under λ-cyhalothrin treatment in *B. gobica* (Figure 5A and Figure 6 and Table 2).

#### 2.3.4. Abamectin Treatment

The *RPL10*, *RPS15* and *GAPDH* were the most stable genes in the Delta Ct, NormFinder/BestKeeper and GeNorm methods, respectively. *RPS20* was the least stable gene in all analyses (Figure 4 and Appendix A). Based on RefFinder, the comprehensive rank order of gene stability was as follows (from the most to least stable): *RPS15*, *RPL10*, *GAPDH*, *RPL32*, *EF1α*, *RPS11*, *EF2*, *GST*, *Ferritin*, *β-actin*, *AK*, *RPL8*, *α-TUB*, *β-TUB*, *RPS20* (Figure 6). GeNorm results (V2/3 < 0.15) determined that a combination of two RGs (*RPS15* and *RPL10*) was a better choice for RT-qPCR normalization under abamectin treatment in *B. gobica* (Figure 5A and Figure 6 and Table 2).

#### 2.3.5. Matrine Treatment

The most stable genes were *RPL10* (Delta Ct/BestKeepe), *RPL8* (NormFinder) and *GST* (GeNorm) in the corresponding analytical method, and the opposite unstable gene was *β-TUB* in all analyses (Figure 4 and Appendix A). These three genes, *RPL10*, *RPL8* and *AK*, were in the top three in the RefFinder analysis (Figure 6). Based on the GeNorm results (V2/3 < 0.15), the *RPL10* and *RPL8* were a better combination for RT-qPCR normalization under matrine treatment in *B. gobica* (Figure 5A and Figure 6 and Table 2).

#### 2.3.6. All Insecticides

The *EF2* (Delta Ct/GeNorm), *GST* (NormFinder) and *β-TUB* (BestKeeper) were the most stable genes with the corresponding four algorithms (Table 3). The *RPS20* was the most unstable gene by all analyses (Figure 6 and Table 3). The RefFinder results indicated that the rank order for gene stability was as follows (from the most to least stable): *EF2*, *RPL32*, *GAPDH*, *EF1α*, *GST*, *α-TUB*, *RPS15*, *RPL8*, *AK*, *β-TUB*, *Ferritin*, *RPL10*, *β-actin*, *RPS11*, *RPS20* (Figure 6). Among them, *EF2* and *RPL32* were considered as the optimal combination RGs for RT-qPCR normalization under five different insecticide stresses in *B. gobica*, as determined from GeNorm results (V2/3 < 0.15) (Figure 5B and Figure 6 and Table 2).

### 2.4. Stability of Candidate RGs under Temperature and Odor Stimulation

#### 2.4.1. Temperature Treatments

*EF1α* was the most stable gene in the Delta Ct and NormFinder analyses, whereas the most stable genes were *RPL8* in GeNorm analysis and *β-actin* in BestKeeper analysis (Table 3). *β-TUB* was the least stable gene in the four analyses (Figure 6, Table 3). The RefFinder results revealed that *EF1α* (1.73) and *β-actin* (2.21) had a lower geometric mean and were the most stable genes (Figure 6 and Appendix A). Based on GeNorm analysis, all pairwise values were less than 0.15. (Figure 5B). Thus, *EF1α* and *β-actin* were the optimal combination for RT-qPCR normalization under different temperature treatments in *B. gobica* (Figure 5B and Figure 6 and Table 2).

#### 2.4.2. Odor Stimulation

*EF1α* was the most stable gene based on the Delta Ct and NormFinder analyses, while *RPL8* was by GeNorm and BestKeeper analyses (Table 3). All four algorithms showed that *RPL32* was the least stable gene. The comprehensive ranking order for gene stability using RefFinder was as follows (from the most to least stable): *EF1α*, *RPL8*, *Ferritin*, *β-actin*, *EF2*, *α-TUB*, *GAPDH*, *RPS15, RPL10, AK*, *β-TUB*, *GST*, *RPS11*, *RPS20, RPL32* (Figure 6). As determined from GeNorm results (V2/3 < 0.15), a combination of *EF1α* and *RPL8* was the optimal combination of RGs for normalization of target gene expression under different odor stimulations in *B. gobica* (Figure 5B and Figure 6 and Table 2).

### 2.5. Stability of Candidate RGs at Developmental Stages and Both Sexes

#### 2.5.1. Developmental Stages

*β-actin* was the most stable gene in the Delta Ct, NormFinder and GeNorm analyses, while *β-TUB* was the most stable gene in BestKeeper analysis (Table 3). All analyses showed that *α-TUB* was the least stable gene (Table 3). The rank order of gene stability by RefFinder was, from the most to least stable genes, as follows: *β-actin*, *EF2*, *RPS15*, *GST*, *RPS11*, *RPS20*, *β-TUB*, *RPL10*, *Ferritin*, *EF1α*, *GAPDH*, *RPL8*, *AK*, *RPL32*, *α-TUB* (Figure 6). In view of the GeNorm results (V2/3 < 0.15), *β-actin* and *EF2* were proposed as the optimal combination for RT-qPCR normalization across developmental stages of *B. gobica* (Figure 5B and Figure 6 and Table 2).

#### 2.5.2. Both Sexes

The GeNorm and NormFinder results showed that *β-actin* and *EF1α* were the most stable genes, whereas the other two analyses found the *RPS13* gene to be most stable (Table 3). *α-TUB* was identified as the most unstable gene in all analyses (Table 3 and Figure 6). From the results of RefFinder (from the most to least stable genes: *β-actin*, *RPS15*, *RPL8*, *Ferritin*, *AK*, *RPS20*, *GST*, *RPL32*, *EF1α*, *RPS11*, *GAPDH*, *RPL10*, *β-TUB*, *EF2*, *α-TUB*) and from GeNorm analysis (Vn/(n + 1) < 0.15), the optimal combination RGs for RT-qPCR normalization in the male and female of *B. gobica* adults were *β-actin* and *RPS15* (Figure 5B and Figure 6 and Table 2).

### 2.6. Comprehensive Ranking Analysis of Candidate RGs for All Experimental Conditions

In all sample conditions, *EF2, GST* and *RPL8* were the most stable genes by GeNorm/Delta Ct, NormFinder and BestKeeper analyses, respectively (Table 3). *RPL32* was the least stable gene (Table 3 and Figure 6). The overall rank order for gene stability determined from RefFinder results was as follows (most to least stable): *EF2*, *RPL8*, *GST*, *Ferritin*, *GAPDH*, *RPS15*, *EF1α*, *RPS11*, *RPL10*, *RPS20*, *AK*, *β-actin*, *β-TUB*, *α-TUB*, *RPL32* (Figure 6). Based on the RefFinder data and GeNorm results (V2/3 < 0.15), *EF2* and *RPL8* are the most suitable internal RGs for normalizing target gene expression by RT-qPCR in *B. gobica* (Figure 5B and Figure 6 and Table 2).

### 2.7. Validation of the Selected Candidate RGs by Target Gene CYP6a1

CYP6a1 has been shown to metabolize certain insecticides (such as imidacloprid, λ-cyhalothrin, etc.) and reduce insecticide tolerance in insect pests [1,29]. Hence, *CYP6a1* of *B. gobica* was used as the target gene to verify expression stability of potential RGs by RT-qPCR at both developmental stages (adults and nymphs) and under imidacloprid stress. As shown in Figure 7A, the transcript level of *CYP6a1* was significantly lower in nymphs than in adults of *B. gobica* by FPKM (fragments per kilobase of exon model per million mapped reads, FPKM) values (Appendix A) of transcriptome data (*p* < 0.05). Further RT-qPCR results showed that the expression levels of *CYP6a1* by using a single most stable RG (*β-actin* or *EF2*) or optimal combination RGs (*β-actin* + *EF2*) for normalizing were consistent with the FPKM of transcriptome data in the adult and nymph stages (Figure 7B). But the *CYP6a1* expression using the corresponding least stable RG (*α-TUB*) for normalization did not differ between nymphs and adults, whereas transcriptome FPKM values in adults were significantly higher than in nymphs (Figure 7A,B). Compared to the control without imidacloprid treatment, the expression of *CYP6a1* based on FPKM values was downregulated in nymphs after imidacloprid treatment (although not significantly) (Figure 7C and Appendix A). For the RT-qPCR results for a single stable RG (*RPL10* or *RPS15*) or a combination of stable RGs (*RPL10* + *RPS15*), normalization showed a similar downregulation trend after imidacloprid treatment (Figure 7D). However, there was a small, but not significant, increase in the *CYP6a1* expression after imidacloprid treatment under the most unstable RG (*β-actin*) normalization (*p* > 0.05) (Figure 7D).

## 3. Discussion

*L. barbarum* is a traditional Chinese herbal medicine because of its rich *L. barbarum* polysaccharide, which is beneficial to the eyes and has anti-aging and regulation of immunity effects [5,6,7]. Chemical pesticides need to be sprayed several times a year to control *B. gobica* during the growth stage of *L. barbarum* L., which has led to increasing insecticide resistance of *B. gobica* in the field [10]. However, the mechanisms underlying the insecticide resistance of *B. gobica* are currently unclear. The screening and validation of internal reference genes for normalization is essential to clarify the differences in the expression of detoxification enzyme genes between sensitive and insecticide-resistant strains of *B. gobica* by RT-qPCR. But there have been no reports of reference genes in *B. gobica*. In this study, 15 commonly used candidate RGs were identified from the transcriptome data of *B. gobica*. Their expression stability was assessed by five algorithms to clarify the appropriate RGs in response to different insecticides for further studies of detoxification enzyme gene function.

The most stable reference gene is different under different experimental conditions in the same insect species [11,15]. Moreover, no single internal RG from any species has been found to be stably expressed under all experimental conditions to date [13,14,15]. Our results of RG screening in *B. gobica* also prove the above points. In general, *β-actin* acts as a stable RG in sex, different tissues and developmental stages of insects [15]. Our experimental results showed that *β-actin* is a stable RG in developmental stages and both sexes, yet was the least stable RG under imidacloprid stress. In addition, many studies have demonstrated that *β-actin* is not stably expressed during these periods in insects [29,30], such as *GAPDH* and *EF1α* for the developmental stages of *Diaphorina citri* [22], and *EF1α* and *RPL5* for the life stages in *Bactericera cockerelli* [21]. The above findings present a normal phenomenon of stable RG selection under the same experimental conditions among diverse insect species [11,15,29]. And sometimes RGs differ greatly in adult and larva stages even in the same tissue [15,30]. For more accurate RT-qPCR quantitative results, two or more RGs are better used for normalization than a single RG [15], based on GeNorm results for the pairwise variation value at V2/3 below the proposed 0.15 cut-off value (Figure 5), which showed that two stable RGs would be a better choice for RT-qPCR normalization compared to a single RG under various experimental conditions.

When insects were treated with different insecticides, their reference genes could not always express stably [15,31]. For instance, *EF1α* was the most stable RG in *D. citri* and *B. odoriphaga* under imidacloprid treatment, while *β-actin* and Ferritin were expressed most stably for chlorpyrifos/chlorfluazuron and beta-cypermethrin treatments, respectively [16]. In our study, the stability of the *B. gobica* RGs was shown to be different in response to different insecticides. For example, *RPL10* + *RPS15*, *RPL10* + AK, *RPL32* + *RPL10* and *RPL10* + *RPL8* were selected as the optimal combination RGs for normalizing gene expression under imidacloprid/abamectin, thiamethoxam, λ-cyhalothrin and matrine, respectively. Our results were similar to those of previous studies [15,16,21]. Other studies have shown that differences in the resistance of aphid populations to pyrethroid affect the stability of their internal reference genes [32]. Therefore, the most stable internal reference genes of insects under each insecticide stress must be identified for better expression quantification of detoxification enzyme genes. In addition, the differences of species in sex and the development stages, as well as the stresses caused by different temperatures and odors, also lead to the differences in the expression stability of RGs [15]. For example, *RPS20* + *Ferritin* in *D. citri* [23] and *RPS13* + *EF1α* in *Neoceratitis asiatica* [11] were the best choices of RGs under different temperature stresses and odor stimulations, respectively. There are a few differences between our results (*EF1α + β-actin* for temperature treatment, *EF1α + RPL8* for odor stimulation) and previous reports under temperature and odor stresses [11,15,23]. Furthermore, studies also indicated that *RPL32* was the most stable RG in *Z. cucurbitae* response to different temperatures [18] and in *N. asiatica* with different tissues [11]. On the contrary, for our results, *RPL32* was the most unstable RG of *B. gobica*. All these results prove that the reference genes must be screened under different experimental conditions of each species.

After the internal reference genes are screened by different analysis methods, these screened RGs need to be verified the accuracy and reliability of PCR quantification with the corresponding target genes [11,15]. In this study, target *CYP6a1* expression in adults of *B. gobica* was significantly higher than in nymphs using the most stable *β-actin* and *EF2* as RGs. However, there were no expression differences of *CYP6a1* between adults and nymphs of *B. gobica* using the most unstable *α-TUB* as the RG (Figure 7). Using validated stable RGs for normalizing to target gene expression is to better ensure the accuracy of RT-qPCR results [11,15,16,17]. Meanwhile, at least two or more reference genes should be selected for gene quantitative normalization [15]. CYP6a1, mainly involved in insecticide metabolism [1,33], was used for RG verification in *B. gobica*. *RPL10* and *RPS15* were the most stable RGs for normalizing to *CYP6a1* expression before and after imidacloprid treatment via RefFinder and GeNorm (V2/3 < 0.15) results. The RT-qPCR results were consistent with the transcriptome data of *B. gobica*. Previous studies have shown that the quantitative results using two or three genes as RGs for target gene normalization were more similar to the transcriptome data in *N. asiatica* [11] and *Anastrepha obliqua* [34]. But if the unstable genes are selected as RGs, RT-qPCR quantitative results will be uncertain and unrealistically presented in insects. As a result, optimization of reference genes is crucial for the accurate normalization of gene expression, especially for identifying the relatively subtle difference.

In conclusion, our study is the first to screen the suitable reference genes under different insecticide stresses and validate the expression levels of target *CYP6a1* between nymph and adult stages and under imidacloprid treatment in *B. gobica*. A single most stable RG and the optimal combination RGs for RT-qPCR normalization were identified in the developmental stages and both sexes and under different temperatures, different odors and five insecticide stresses in *B. gobica* based on five algorithms (Table 2). These results are helpful to further elucidate the resistance mechanisms of *B. gobica* to these single insecticides (including imidacloprid, thiamethoxam, λ-cyhalothrin, abamectin and matrine) or to neonicotinoid insecticides (such as imidacloprid, thiamethoxam, etc.) or pyrethroid pesticides (λ-cyhalothrin, etc.). Additionally, our results further emphasize that the most suitable reference genes should be screened and validated under different experimental conditions to ensure the accuracy of gene quantification.

## 4. Materials and Methods

### 4.1. Insect Rearing

*B. gobica’s* nymphs (mainly fifth instar nymphs) and adults (male and female) were collected from an organic wolfberry producing area in Ningxia, China, during the wolfberry harvest on 20 July 2021. Then approximately 5000 nymphs and 1000 adults were maintained in the institute of Medicinal Plant Development, Chinese Academy of Medical Sciences and Peking Union Medical College. These nymphs and adults were raised on organic goji berry seedlings by an artificial climate cabinet (PXZ-430B, Ningbo Jiangnan Instrument Factory, China, Ningbo) with a 14 Light (L):10 Dark (D) photoperiod at 25 ± 2 °C and 60 ± 10% relative humidity. *B. gobica’s* nymphs and adults were reared under indoor conditions for about 2 years. Then they were treated and collected as experimental material for this study on 10–15 April 2023, the detailed processing methods were as follows.

### 4.2. Collection of Samples, RNA Extraction and cDNA Synthesis under Different Experimental Conditions

The fifth-instar nymphs and adults moulting for 2 days of *B. gobica* were used in the experiments. Each experiment was set as three biological replicates. The collected methods used in this study were similar to those of previous studies [11]. The detailed information of sample collection under different experimental conditions is as follows. (1) Different insecticide treatments. The *B. gobica* nymphs were treated with different concentrations (including 0.1 mg/L, 0.01 mg/L, and 0.001 mg/L) of different insecticides (Imidacloprid, Thiamethoxam, λ-cyhalothrin, Abamectin and Matrine) by the leaf impregnation method for 24 h. Thirty individuals were used for each concentration, with three biological replicates. The control group was 30 *B. gobica* nymphs treated with water for 24 h, with three biological replicates. Then these nymphs were collected in different microcentrifuge tubes (1.5 mL, no RNase). (2) Temperature treatments. The *B. gobica* nymphs were exposed to three different temperatures (including 10 °C, 20 °C and 30 °C) for 2 h. 30 individuals were collected for each temperature, with three biological replicates. (3) Odor stimulation. The *B. gobica* adults were exposed to three odor compounds (1 mM (E)-2-Hexenal, 1 mM Linalool and 1 mM D-Limonene) for 2 h (h), respectively. Forty individuals were collected for each odor compound, with three biological replicates. The other 40 adults were unstimulated with odor compounds as a control group, with three biological replicates. The stimulated and unstimulated adults were collected separately. (4) Different developmental stages. Two hundred eggs (3 days old), 30 fifth-instar nymphs and 40 adults (a mixture of males and females, male: female = 1:1) were collected separately, with three biological replicates. (5) Both sexes. Forty adult male and forty adult female individual *B. gobica* were collected, with three biological replicates. All samples collected above were quickly frozen in liquid nitrogen, then stored at −80 °C.

The techniques and methods used in this study were similar to those of our previous studies [32,34]. The RNA and first-strand cDNA of these above collected samples were extracted and synthesized using the Invitrogen TRIzol Reagent (Invitrogen, Carlsbad, CA, USA) and the primeScriptTM RT reagent Kit with gDNA Eraser (Takara, Dalian, China), respectively. The total RNA quantity and quality were determined by a NanoDrop 2000 spectrophotometer (NanoDrop, Wilmington, DE, USA). Then 1 μg of total RNA was used to synthesize the cDNA.

### 4.3. Selection, Primer Design and RT-qPCR Analysis of Reference Genes (RGs) in B. gobica

Fifteen genes, including *β-actin, EF1α, EF2, GAPDH, α-TUB, β-TUB, Ferritin, GST, AK, RPL8, RPL10, RPL32, RPS11, RPS15 and RPS20,* identified from transcriptome sequencing (accession numbers: SRR27779037~SRR27779040, SRR27779044~SRR27779048 in NCBI) of *B. gobica* adults and nymphs with FPKM value greater than 100, were selected as candidate RGs (Appendix A). These genes had been reported as RGs for RT-qPCR analysis in other Hemiptera insects [15,21,22,23]. The specific primers of the selected genes were designed using NCBI Primer-BLAST with exon junction span (Primer must span an exon–exon junction) (https://www.ncbi.nlm.nih.gov/tools/primer-blast/index.cgi?LINK_LOC=BlastHome, accessed on 23 March 2023) and listed in Table 1.

The synthetic cDNA was diluted in a 5-fold series (1/5, 1/25, 1/125, 1/625 and 1/3125), and the dilutions were used to generate a standard curve. The RT-qPCR amplification efficiency (E) (E = (10[−1/slope] − 1) × 100) and correlation coefficient (*R*^2^) of each primer were calculated and analyzed automatically by Bio-Rad CFX 3.0 software. The specificity of RT-qPCR primers was confirmed by the melting curve and sequencing of the RT-qPCR products (Figure 2) [11,35].

The techniques and methods of RT-qPCR used in this study were similar to those of our previous studies [11,35]. Each RT-qPCR reaction was conducted in a 20 μL reaction: 10 μL of SYBR ^®^ Premix Ex TaqTM (TliRNase H Plus) (Takara, China), 0.75 μL of each primer (10 μM), 1 μL of sample cDNA, and 7.5 μL of sterilized ddH_2_O. The reactions were performed on a StepOne thermocycler (Bio-Rad CFX, Hercules, CA, USA) using the two-step method (run as follows: 94 °C for 2 min, followed by 40 cycles of 94 °C for 15 s, 60 °C for 30 s, 60 °C for 1 min) and were analyzed with a melting curve analysis program (heated to 95 °C for 30 s and cooled to 60 °C for 15 s). The Ct values were obtained from RT-qPCR results analyzed by the Bio-Rad CFX 3.0 software.

### 4.4. Determining the Expression Stability of Candidate RGs

The RT-qPCR data for five experimental conditions were analyzed independently. Each sample for one RG was set as three biological and three technical replicates by RT-qPCR analysis. Four algorithms (Delta Ct method [24], GeNorm [25], NormFinder [26], BestKeeper [27]) were used to evaluate the expression stability of 15 candidate RGs. The comprehensive rank was calculated by RefFinder [28], which could unify and merge the above four algorithms. A suitable number of RGs for RT-qPCR normalization was determined by GeNorm analysis [15]. Analyses of these algorithms used in this study were similar to those of previous studies [11,12,13,14,15,16].

### 4.5. Validation of the Selected RGs by Target Gene CYP6a1

To confirm the reliability of the potential reference genes, the relative expression of target *CYP6a1* was measured by RT-qPCR between adults and nymphs of *B. gobica* and under imidacloprid stress. The *CYP6a1* was identified from transcriptome sequencing of *B. gobica* nymphs and adults, whose details (including sequence analysis, primer evaluation) were listed in Appendix A and Appendix A. At present, the clean data of *B. gobica* transcriptome have been uploaded to the SRA database to the National Center for Biotechnology Information (NCBI); their corresponding accession numbers are listed in detail in Table 4.

The *CYP6a1* expression was normalized with the most stable and least stable RGs obtained from the comprehensive analysis of GeNorm and RefFinder. The study method of the *CYP6a1* by RT-qPCR was described in detail in Section 4.3. Each experimental sample was set as three biological and three technical replicates in RT-qPCR analysis. The relative expression level of target *CYP6a1* was calculated according to the 2^−∆∆ Ct^ method [36]. The FPKM value of *CYP6a1* in transcriptome data was used as a reference for its expression pattern (Appendix A).

### 4.6. Statistical Analysis

All statistical comparison was determined using SPSS 22.0 software (SPSS Inc., Chicago, IL, USA). Data multiple comparison was assessed by ANOVA following Tukey’s HSD multiple comparison test (*p* < 0.05). The statistical significance of the difference between two treatments was analyzed using a pairwise Student’s *t* test (*p* < 0.05) [11].

## Figures and Tables

**Figure 1 ijms-25-02434-f001:**
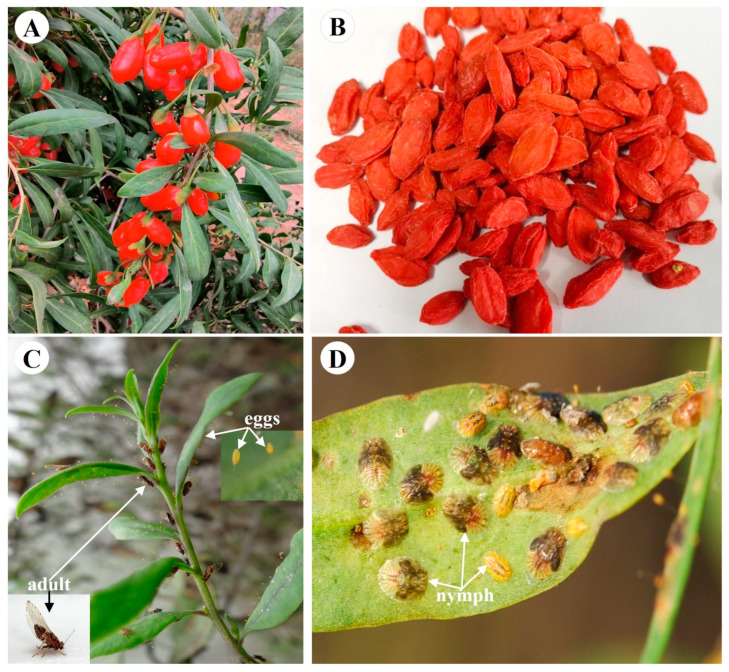
*Lycium barbarum* L. and its major pest *Bactericera gobica*. (**A**) Fresh ripe fruits of *L. barbarum* L. (**B**) Dried ripe fruits of *L. barbarum* L. (**C**) *B. gobica* adults laying eggs on young leaves of *L. barbarum* L. (**D**) *B. gobica* nymphs feeding on leaf of *L. barbarum* L.

**Figure 2 ijms-25-02434-f002:**
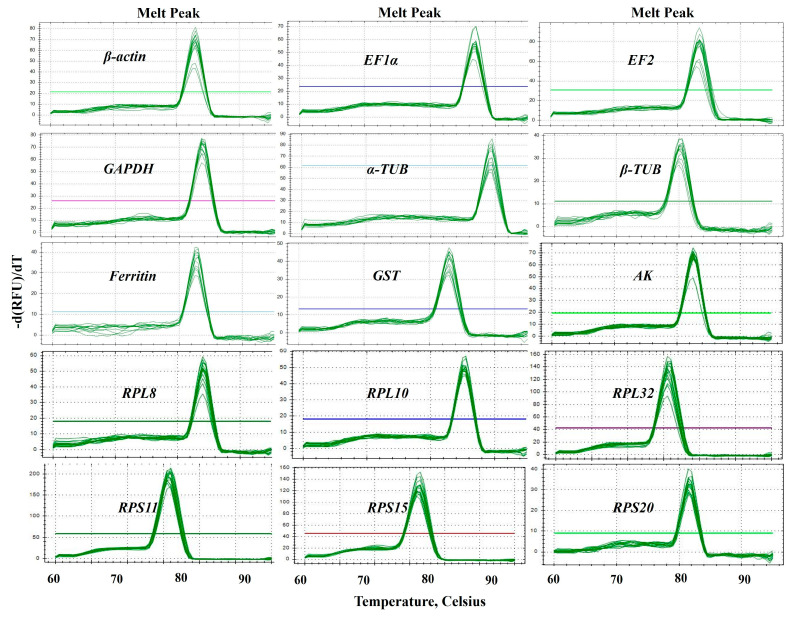
Specificity of primer pairs for RT-qPCR amplification in *B. gobica*. The melt peaks of primers for RT-qPCR amplification of 15 candidate RGs, including *β-actin*, *EF1α*, *EF2*, *GAPDH*, *α-TUB*, *β-TUB*, *Ferritin*, *GST*, *AK*, *RPL8*, *RPL10*, *RPL32*, *RPS11*, *RPS15* and *RPS20*. The straight lines with different colors indicated the template-free negative controls for different genes.

**Figure 3 ijms-25-02434-f003:**
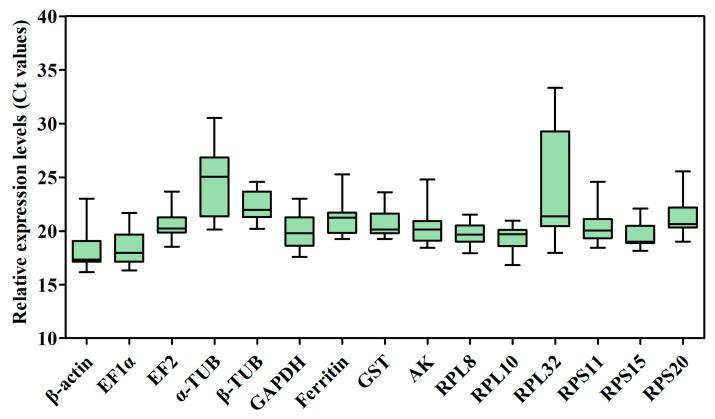
Candidate RG expression profiles in *B. gobica*. The expression data are presented as mean Ct values of candidate RGs across all samples under different experimental conditions. Whiskers represent the maximum and minimum values. The lower and upper borders of boxes represent the 25th and 75th percentiles, respectively. The line across the box indicates the median Ct value.

**Figure 4 ijms-25-02434-f004:**
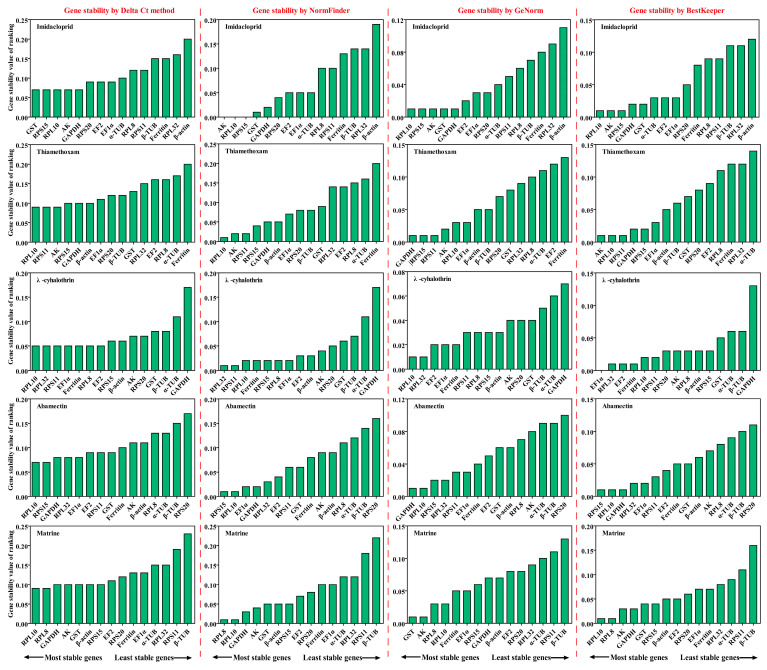
Expression stability ranking of 15 candidate RGs in five different insecticides was calculated using Delta Ct, GeNorm, NormFinder and BestKeeper. The five different insecticides included imidacloprid, thiamethoxam, λ-cyhalothrin, abamectin, matrine. The expression stability (standard value, SV) is listed (the lower, the most stable).

**Figure 5 ijms-25-02434-f005:**
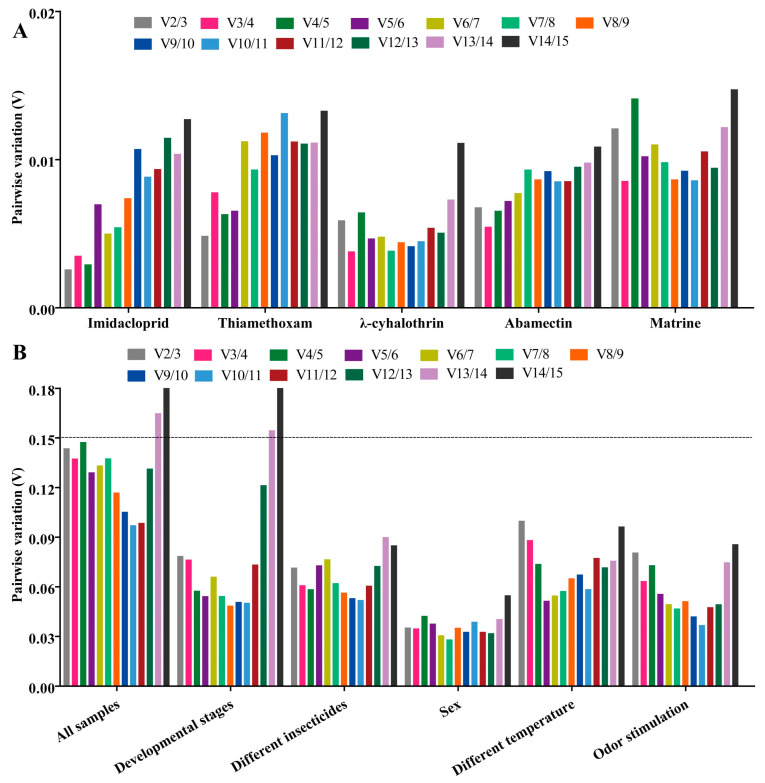
GeNorm analysis of paired variation (V) values of 15 candidate RGs. Vn/Vn + 1 values are used to determine the optimal number of reference genes. The cut-off value to determine the optimal number of reference genes for RT-qPCR normalization is 0.15. (**A**) Genorm analysis of paired variation (V) values of 15 candidate RGs under five different insecticide treatments. (**B**) Genorm analysis of paired variation (V) values of 15 candidate RGs under different experimental conditions.

**Figure 6 ijms-25-02434-f006:**
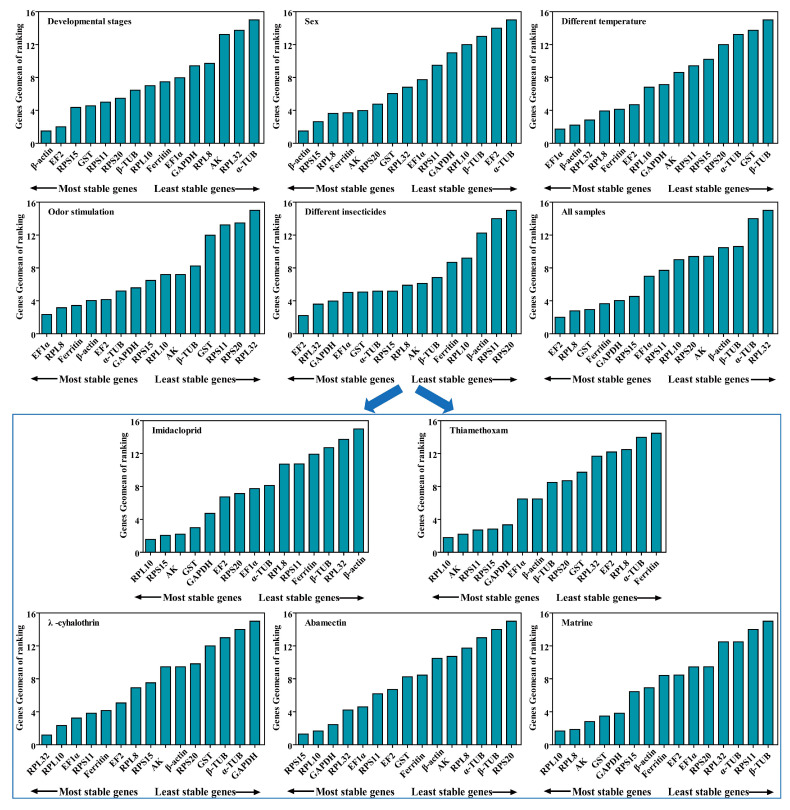
Stability of 15 candidate RGs in *B. gobica* under five experimental conditions by RefFinder analysis. In a RefFinder analysis, increasing Geomean values corresponds to decreasing gene expression stability. The Geomean values for the following *B. gobica* samples are presented: Odor stimulation: samples treated with different odor compounds ((*E*)-2-Hexenal, Linalool and D-Limonene); insecticide treatments: samples treated with five different insecticides (Imidacloprid, Thiamethoxam, λ-cyhalothrin, Abamectin and Matrine); Temperature: samples treated with different temperatures (10 °C, 20 °C and 30 °C); Developmental stages: samples for three developmental stages (eggs, nymphs and adults); Sex: samples for male adults and female adults; All samples: all samples for all treatments under different experimental conditions. The details of 15 candidate RGs are listed in Table 1. The most stable genes are listed on the left, while the least stable genes are listed on the right.

**Figure 7 ijms-25-02434-f007:**
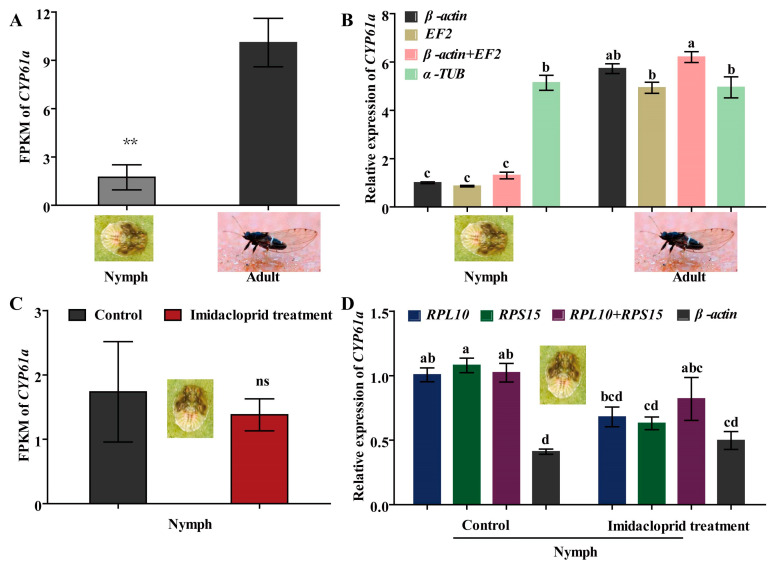
The expression profiles of *CYP6a1* in adult and nymph stages and under imidacloprid stress in *B. gobica*. (**A**) The expression levels of *CYP6a1* in the adult and nymph stages presented by FPKM value of transcriptome data. FPKM: fragments per kilobase of exon model per million mapped reads. Asterisks (**) indicate a significant difference was found between nymph and adult (mean ± SE, *n* = 3, Student’s *t* test, ** *p* < 0.01). (**B**) The expression levels of *CYP6a1* in the adult and nymph stages by RT-qPCR. The expression levels of *CYP6a1* in nymphs were used for normalization. Namely, the relative expression levels were presented as fold changes relative to the transcript levels of *CYP6a1* in nymphs. *β-actin*, *EF2* and *β-actin* + *EF2* were used as the one or two most stable reference genes. *α-TUB* was used as the least stable reference gene. Data are represented as the mean ± SE. Bars labelled with different letters are significantly different (*p* < 0.05, ANOVA followed by Tukey’s HSD multiple comparison test, *n* = 3). (**C**) The expression levels of *CYP6a1* in nymphs before/after imidacloprid treatment presented by FPKM value of transcriptome data. Control shows the nymphs without imidacloprid treatment. “ns” indicates no difference was between control and imidacloprid treatment (mean ± SE, *n* = 3, Student’s *t* test, ns *p* < 0.01). (**D**) The expression levels of *CYP61a* in nymphs normalized to the top two stable genes (*RPL10* and *RPS15*) and an unstable gene (*β-actin*) by RT-qPCR. The expression level of *CYP6a1* in nymphs before imidacloprid treatment was used for normalization. Bars labelled with different letters are significantly different (mean ± SE, *n* = 3, Tukey’s HSD, *p* < 0.05).

**Table 1 ijms-25-02434-t001:** Primer sequences and amplification sizes of the selected RGs and target gene.

Gene Symbol	Gene Name	Primer Sequence (from 5′ to 3′)	Amplicon Size (bp)	E (%)	*R* ^2^	Slope
Reference genes:						
*β-actin*	Beta actin	F: GGTGTCATGGTGGGTATGGG R: CAGGATGGGGTGCTCTTCTG	150	98.3	0.995	−3.365
*EF1α*	Elongation factor 1 alpha	F: CGGAAAAACCACCGAGGAGA R: TTGATGACTCCCACAGCCAC	144	98.5	0.999	−3.357
*EF2*	Elongation factor 2	F: GTACAGGCCCCAACCTTCTC R: GGACAGGACACCCTCTTTGG	149	92.7	0.997	−3.510
*Ferritin*	Ferritin	F: GCAGAACTCTTGCTCGTCCT R: AGGGTGTACTGGTAGCTGGT	105	96.2	0.989	−3.416
*GAPDH*	Glyceraldehyde-3-phosphate dehydrogenase	F: ATGGAAAGTTCAAGGGAGACG R: AACATACTCAGCTCCAGACTTG	148	97.3	0.998	−3.388
*α-TUB*	Alpha tubulin	F: AAGTCTCCACCTCCGTCGTA R: TCAGGTTGGTGTAGGTGGGA	172	90.5	0.997	−3.572
*β-TUB*	Beta-tubulin	F: CTTGTATCCCTCACCATGTCC R: TCTCAGGTCAGCATTCAGTTG	99	93.5	0.997	−3.488
*AK*	Arginine kinase	F: GGCCACAGAAAGCAAATCCC R: GGATTCAGCATCAGGAGCGT	131	91.3	0.992	−3.551
*GST*	Glutathione S-transferase	F: CCCAAGCAGAGGGGAATTGT R: AGGCGAGGAAAGTGTTGAGG	122	98.7	0.996	−3.355
*RPL8*	Ribosomal protein L8	F: GAGCTGTTCATTGCCCCAGA R: ATTCCTCCCACAGGCATCAC	198	92.2	0.992	−3.523
*RPL10*	Ribosomal protein L10	F: GTATGCGTGGTGCCTATGGT R: TCCGGGGAACTTGAACTTGG	129	98.3	0.993	−3.364
*RPL32*	Ribosomal protein L32	F: AGGAAAGTCTTCATCCGTCATC R: TTCTTGGCACTACCGTAACC	116	91.5	0.998	−3.544
*RPS11*	Ribosomal protein S11	F: CTTTCCAAAAGCAGCCGACA R: TGACGTTACCAGTGAAGGGG	142	93.2	0.999	−3.497
*RPS15*	Ribosomal protein S15	F: GGACGAATAACCTGCTACCATC R: CCTCCCAAGTGTTTTCTCCTG	92	92.9	0.999	−3.505
*RPS20*	Ribosomal protein S20	F: CCACCACCCACAAAGGATCT R: TGATGGGACCCTTTGGCTTC	92	104.4	0.997	−3.220
Target gene:						
*CYP6a1*	cytochrome P450 CYP6a1	F: AGGAATGATCTCTTGCAGACG R: GTGGTCAATGCGGATGTTTC	149	97.5	0.996	−3.384

Note: F: forward primer, R: reverse primer. “E” indicates the primer efficiency for RT-qPCR amplification (E calculated by the standard curve method). “*R*^2^” indicates regression coefficient of the standard curve.

**Table 2 ijms-25-02434-t002:** Single most stable and optimal combination RGs in *B. gobica* under different experimental conditions.

Experimental Conditions	Single Most Stable Reference Genes	Optimal Combination Reference Genes
Imidacloprid	*RPL10*	*RPL10* + *RPS15*
Thiamethoxam	*RPL10*	*RPL10* + *AK*
λ-cyhalothrin	*RPL32*	*RPL32* + *RPL10*
Abamectin	*RPS15*	*RPS15* + *RPL10*
Matrine	*RPL10*	*RPL10* + *RPL8*
Different insecticides	*EF2*	*EF2* + *RPL32*
Different temperature	*EF1α*	*EF1α* +*β-actin*
Odor stimulation	*EF1α*	*EF1α* + *RPL8*
Developmental stages	*β-actin*	*β-actin* + *EF2*
Both sexes	*β-actin*	*β-actin* + *RPS15*
All samples	*EF2*	*EF2 +* *RPL8*

**Table 3 ijms-25-02434-t003:** Expression stability ranking of 15 candidate RGs in all samples was calculated using Delta Ct, GeNorm, NormFinder and BestKeeper. The average expression stability (M value) is listed by Delta Ct, GeNorm and NormFinder analyses, and standard deviation (SD) and coefficient of variation (CV) were given by BestKeeper analysis (the lower, the most stable). The stability decreases from top to bottom.

Rank	Delta Ct Method		NormFinder		GeNorm		BestKeeper		
	Gene	M	Gene	M	Gene	M	Gene	SD	CV
Different insecticides									
1	*EF2*	0.62	*GST*	0.28	*EF2*	0.16	*β-TUB*	0.59	2.45
2	*RPL32*	0.65	*RPL32*	0.29	*GAPDH*	0.16	*RPL8*	0.82	4.05
3	*RPS15*	0.67	*RPS15*	0.29	*α-TUB*	0.20	*α-TUB*	0.84	3.11
4	*EF1α*	0.67	*EF2*	0.32	*EF1α*	0.24	*GAPDH*	0.87	3.99
5	*AK*	0.67	*AK*	0.33	*RPL8*	0.27	*EF1α*	0.95	4.65
6	*GST*	0.68	*RPL10*	0.38	*RPL32*	0.34	*EF2*	0.98	4.42
7	*GAPDH*	0.69	*Ferritin*	0.41	*AK*	0.40	*RPL32*	1.12	5.78
8	*α-TUB*	0.70	*EF1α*	0.42	*RPS15*	0.44	*AK*	1.22	5.77
9	*RPL10*	0.72	*GAPDH*	0.47	*Ferritin*	0.47	*Ferritin*	1.26	5.55
10	*Ferritin*	0.72	*α-TUB*	0.50	*GST*	0.50	*RPS15*	1.32	6.64
11	*RPL8*	0.75	*RPL8*	0.57	*RPL10*	0.53	*GST*	1.45	6.70
12	*β-actin*	0.83	*β-actin*	0.59	*β-actin*	0.57	*RPL10*	1.55	8.50
13	*β-TUB*	1.15	*β-TUB*	1.09	*β-TUB*	0.63	*β-actin*	1.77	8.57
14	*RPS11*	1.24	*RPS11*	1.16	*RPS11*	0.73	*RPS11*	2.18	10.45
15	*RPS20*	1.32	*RPS20*	1.26	*RPS20*	0.81	*RPS20*	2.39	10.82
Different temperatures									
1	*EF1α*	0.60	*EF1α*	0.11	*RPL8*	0.22	*β-actin*	0.01	0.08
2	*β-actin*	0.64	*Ferritin*	0.14	*RPL32*	0.22	*EF2*	0.22	1.11
3	*Ferritin*	0.64	*β-actin*	0.21	*EF1α*	0.29	*EF1α*	0.25	1.42
4	*RPL32*	0.66	*RPL32*	0.30	*β-actin*	0.34	*RPL32*	0.28	1.33
5	*RPL8*	0.68	*RPL10*	0.34	*EF2*	0.37	*AK*	0.32	1.52
6	*RPL10*	0.71	*RPL8*	0.34	*GAPDH*	0.37	*GAPDH*	0.35	1.80
7	*EF2*	0.71	*EF2*	0.47	*Ferritin*	0.39	*Ferritin*	0.36	1.69
8	*GAPDH*	0.73	*RPS11*	0.50	*RPL10*	0.42	*RPL8*	0.37	1.83
9	*RPS11*	0.80	*GAPDH*	0.51	*RPS15*	0.46	*RPL10*	0.42	2.23
10	*RPS15*	0.87	*AK*	0.66	*AK*	0.52	*RPS11*	0.44	2.08
11	*AK*	0.88	*RPS15*	0.72	*RPS11*	0.55	*RPS15*	0.47	2.50
12	*RPS20*	0.96	*RPS20*	0.79	*RPS20*	0.63	*RPS20*	0.80	3.72
13	*α-TUB*	1.02	*α-TUB*	0.88	*α-TUB*	0.68	*GST*	0.81	3.96
14	*GST*	1.19	*GST*	1.13	*GST*	0.74	*α-TUB*	0.89	3.50
15	*β-TUB*	1.48	*β-TUB*	1.43	*β-TUB*	0.84	*β-TUB*	1.30	5.94
Odor stimulation									
1	*EF1α*	0.49	*EF1α*	0.14	*RPL8*	0.15	*RPL8*	0.10	0.50
2	*Ferritin*	0.52	*Ferritin*	0.18	*β-actin*	0.15	*β-actin*	0.12	0.69
3	*α-TUB*	0.53	*α-TUB*	0.22	*GAPDH*	0.22	*EF2*	0.14	0.68
4	*RPS15*	0.53	*RPS15*	0.24	*EF2*	0.25	*GAPDH*	0.16	0.87
5	*EF2*	0.54	*EF2*	0.25	*EF1α*	0.31	*Ferritin*	0.26	1.31
6	*β-TUB*	0.56	*AK*	0.27	*RPL10*	0.33	*EF1α*	0.27	1.58
7	*AK*	0.56	*β-TUB*	0.28	*Ferritin*	0.35	*RPL10*	0.31	1.54
8	*RPL10*	0.58	*RPL10*	0.32	*AK*	0.37	*AK*	0.32	1.71
9	*GAPDH*	0.59	*GAPDH*	0.40	*α-TUB*	0.40	*α-TUB*	0.50	2.39
10	*RPL8*	0.62	*RPL8*	0.47	*RPS15*	0.41	*β-TUB*	0.50	2.27
11	*β-actin*	0.64	*GST*	0.47	*β-TUB*	0.43	*RPS15*	0.52	2.57
12	*GST*	0.66	*β-actin*	0.48	*GST*	0.46	*RPS20*	0.64	3.20
13	*RPS11*	0.74	*RPS11*	0.60	*RPS11*	0.49	*GST*	0.70	3.38
14	*RPS20*	1.14	*RPS20*	1.12	*RPS20*	0.57	*RPS11*	0.82	4.07
15	*RPL32*	1.30	*RPL32*	1.27	*RPL32*	0.67	*RPL32*	1.38	4.90
Developmental stages									
1	*β-actin*	0.81	*β-actin*	0.05	*β-actin*	0.09	*β-TUB*	0.41	1.94
2	*EF2*	0.82	*EF2*	0.05	*EF2*	0.09	*RPL10*	0.47	2.35
3	*RPS15*	0.83	*GST*	0.08	*RPS15*	0.19	*RPS11*	0.60	2.96
4	*GST*	0.87	*RPS15*	0.10	*GST*	0.26	*EF2*	0.69	3.40
5	*RPS11*	0.87	*RPS20*	0.15	*RPS20*	0.28	*β-actin*	0.76	4.42
6	*RPS20*	0.89	*GAPDH*	0.24	*RPS11*	0.31	*RPS20*	0.80	3.81
7	*Ferritin*	0.94	*RPS11*	0.33	*Ferritin*	0.36	*EF1α*	0.81	4.63
8	*EF1α*	0.96	*Ferritin*	0.38	*EF1α*	0.39	*Ferritin*	0.83	4.02
9	*RPL8*	0.98	*EF1α*	0.49	*RPL8*	0.41	*GST*	0.87	4.24
10	*GAPDH*	0.99	*RPL8*	0.49	*RPL10*	0.44	*RPS15*	0.91	4.50
11	*RPL10*	0.99	*RPL10*	0.57	*GAPDH*	0.47	*RPL8*	0.92	4.69
12	*β-TUB*	1.26	*β-TUB*	0.97	*β-TUB*	0.55	*GAPDH*	1.13	5.70
13	*AK*	1.74	*AK*	1.40	*AK*	0.71	*RPL32*	1.23	3.92
14	*RPL32*	2.40	*RPL32*	2.33	*RPL32*	0.92	*AK*	2.04	9.54
15	*α-TUB*	3.39	*α-TUB*	3.36	*α-TUB*	1.25	*α-TUB*	3.40	13.55
Both sexes									
1	*β-actin*	0.31	*β-actin*	0.04	*AK*	0.06	*β-actin*	0.02	0.09
2	*RPS15*	0.31	*RPS15*	0.04	*RPL8*	0.06	*Ferritin*	0.08	0.43
3	*Ferritin*	0.33	*Ferritin*	0.08	*GST*	0.09	*RPS15*	0.08	0.41
4	*RPS20*	0.35	*RPS20*	0.14	*EF1α*	0.12	*RPS20*	0.15	0.73
5	*RPL8*	0.37	*RPL32*	0.20	*β-actin*	0.16	*AK*	0.16	0.83
6	*RPL32*	0.37	*AK*	0.21	*RPS15*	0.19	*RPL8*	0.16	0.86
7	*AK*	0.37	*RPL8*	0.22	*Ferritin*	0.20	*GST*	0.17	0.87
8	*GST*	0.38	*GST*	0.23	*RPS20*	0.21	*EF1α*	0.20	1.15
9	*RPS11*	0.41	*RPS11*	0.27	*RPL32*	0.24	*RPL32*	0.20	0.87
10	*EF1α*	0.43	*EF1α*	0.31	*RPS11*	0.26	*RPS11*	0.20	1.05
11	*GAPDH*	0.47	*GAPDH*	0.39	*GAPDH*	0.30	*GAPDH*	0.35	1.93
12	*RPL10*	0.47	*RPL10*	0.39	*RPL10*	0.32	*β-TUB*	0.36	1.68
13	*β-TUB*	0.51	*β-TUB*	0.43	*β-TUB*	0.34	*RPL10*	0.36	1.80
14	*EF2*	0.65	*EF2*	0.58	*EF2*	0.38	*EF2*	0.46	2.39
15	*α-TUB*	0.84	*α-TUB*	0.81	*α-TUB*	0.44	*α-TUB*	0.65	3.07
All samples									
1	*EF2*	1.41	*GST*	0.44	*EF2*	0.53	*RPL8*	0.85	4.31
2	*Ferritin*	1.42	*RPL8*	0.54	*GAPDH*	0.53	*RPS15*	0.90	4.60
3	*GST*	1.47	*RPS15*	0.55	*Ferritin*	0.57	*RPL10*	1.03	5.38
4	*GAPDH*	1.48	*EF2*	0.57	*EF1α*	0.61	*EF2*	1.04	5.01
5	*RPL8*	1.50	*Ferritin*	0.60	*GST*	0.73	*GST*	1.07	5.13
6	*EF1α*	1.57	*GAPDH*	0.74	*RPL8*	0.78	*Ferritin*	1.12	5.26
7	*RPS15*	1.61	*RPS11*	0.79	*β-actin*	0.85	*RPS11*	1.19	5.80
8	*RPS11*	1.63	*AK*	0.79	*RPS20*	0.95	*β-TUB*	1.24	5.52
9	*RPS20*	1.68	*RPS20*	0.94	*RPS11*	1.01	*AK*	1.25	6.11
10	*AK*	1.68	*EF1α*	1.01	*RPS15*	1.05	*EF1α*	1.32	7.16
11	*β-actin*	1.77	*β-TUB*	1.26	*AK*	1.09	*GAPDH*	1.32	6.58
12	*β-TUB*	1.84	*β-actin*	1.31	*β-TUB*	1.14	*RPS20*	1.35	6.33
13	*RPL10*	2.13	*RPL10*	1.44	*RPL10*	1.25	*β-actin*	1.55	8.45
14	*α-TUB*	2.73	*α-TUB*	2.44	*α-TUB*	1.42	*α-TUB*	2.59	10.50
15	*RPL32*	5.47	*RPL32*	5.39	*RPL32*	1.96	*RPL32*	4.27	18.04

**Table 4 ijms-25-02434-t004:** Accession numbers of the clean data of *B. gobica* transcriptome in NCBI.

Clean Data Name	Accession Numbers in NCBI
Nymphs without imidacloprid treatment 1	SRR27779040
Nymphs without imidacloprid treatment 2	SRR27779039
Nymphs without imidacloprid treatment 3	SRR27779038
Nymphs with imidacloprid treatment 1	SRR27779037
Nymphs with imidacloprid treatment 2	SRR27779046
Nymphs with imidacloprid treatment 3	SRR27779045
Adults without imidacloprid treatment 1	SRR27779048
Adults without imidacloprid treatment 2	SRR27779047
Adults without imidacloprid treatment 3	SRR27779044

## Data Availability

All data in this study will be available from the corresponding author upon reasonable request.

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
