# Peer review of "Selection and Validation of Reference Genes for Gene Expression in Bactericera gobica Loginova under Different Insecticide Stresses"

_ijms, 2024, doi:10.3390/ijms25042434_

Round 1
Reviewer 1 Report
Comments and Suggestions for Authors
‘Selection and Validation of Reference Genes for Gene Expression in Bactericera gobica Loginova Under different Insecticide Stresses’ by Wei et al assesses 15 B. gobica genes for their expression stability across different life stages and treatments. I agree with the authors that this is a critical step to ensure accurate gene quantification for a range of future experiments. While the experiments and analyses seem comprehensive, the methods did not clearly explain the number of samples for each, and so the results section also caused me uncertainty as to what, in some cases, was being measured. I suspect some of my issues may be easily fixed by rewording, but as it stands, I’m concerned that there is insufficient biological replication.
Below are some specific comments:
Abstract
Line 11: replace ‘dissecting’ with ‘investigating’
Line 16: replace ‘other four’ with ‘four other’
Lines 17-19: for this, semicolons are needed for clarity, ie ‘Our results indicated that the RGs RPL10 + RPS15 for Imidacloprid and Abamectin; RPL10 + AK for Thiamethoxam; RPL32 + RPL10 for λ-cyhalothrin; RPL10 + RPL8 for Matrine; and EF2 + RPL32 under different insecticide stresses, were selected as the most suitable RGs for RT-qPCR normalization.’
Line 22: respectively
Introduction
Line 32: ‘more’ frequent or ‘higher’
Line 38-39: change to ‘Its dried ripe fruits (also named wolfberry, goji berry, gouqizi) (Figure 1B) are used for medicine and food, and have the effect of anti-aging…’
Line 40-41: ‘(leaves, flowers, fruits, et al)’ remove et al or replace with etc.
Line 44-45: change to ‘which depletes the nutrients of L. barbarum L. and causes the leaves to become yellow and wilt in large areas, resulting in annual yield losses of 20%- 50%’
Line 57-58: change to ‘RT-qPCR has been widely used in molecular biology, including research on insecticide resistance mechanisms, insect-host plant interactions and insect adaptation.’
Line 59-61: change to ‘RT-qPCR data analysis must include the most stable reference genes (RGs) to normalize target gene expression, to optimise the accuracy of quantitative results’
Line 62: replace ‘from’ with ‘in’
Line 64-71: change to ‘Even in the same insect species, the most stable RGs had a significant difference under different insecticide stresses. For instance in Bradysia odoriphaga, the most suitable RGs were Elongation factor 1α (EF1α), beta actin (β-actin) and Ribosomal protein L10 (RPL10) in response to imidacloprid, chlorfluazuron and phoxim, respectively [16]. In addition, β-actin is often used as a stable RG in insects, but is not equally or stably expressed in the two closely related species Bactrocera dorsalis and Zeugodacus (formerly Bactrocera) cucurbitae under the same conditions [17, 18]. Therefore, screening of stable RGs for each experimental condition is a must to ensure the accuracy of RT-qPCR results.’
Line 77-80: change to ‘In contrast, there are relatively few studies on RG screening and target gene functions involved in insecticide resistance and olfactory perception in Chinese herbal pests [11]. To our knowledge, there are no reports identifying the most suitable RGs under different insecticide stresses in B. gobica.’
Line 93-94: change to ‘The proposed research provides the most stable reference genes of B. gobica across different insecticide treatments …’
Results
Line 122-127: Please expand this paragraph to make it clearer. Ie, both temperature and sex had zero s.e. for b-actin but only temp is mentioned.
Line 135: remove ‘respectively’
Line 164: ‘were the most stable genes,’
Line 167-169: Here the values are included whereas previously just the gene ranking. Unless there is a reason to include the values here, just use the gene names. Also, Table S4 doesn’t show those values for λ-cyhalothrin treatment alone.
Line 171-172: ‘under color induction’? Do you mean under λ-cyhalothrin treatment?
Line 180-181: ‘under starvation-refeeding treatment in B. gobica’? Do you mean under Abamectin treatment?
Fig 6: The legend doesn’t match the figure. There is no labelling of A-I, and it mentions color induction and starvation-refeeding which are not part of the paper.
Line 248: replace bank with rank
Line 252: ‘RT-qPCR normalization across developmental stages’
Line 277: nymphs
Line 278: adults
Lines 281-283: change to ‘But CYP6a1 expression using the corresponding least stable RG (α-TUB) for normalization did not differ between nymphs and adults, whereas transcriptome FPKM values in adults was significantly higher than in nymphs.’
Line2 284-286: change to ‘Compared to the control without imidacloprid treatment, the expression of CYP6a1 based on FPKM values was downregulated in nymphs after imidacloprid treatment (although not significantly)’
Discussion
Line 318-319: change to ‘In this study, 15 commonly used candidate RGs were identified from the transcriptome data of B. gobica’
Line 320: replace respond with response.
Line 343: replace respond with response
Line 351” replace quantify with quantification
Line 379: replace identification with identifying
Methods
Section 4.2: I’m confused by the methods in terms of replication. Please be more specific. It seems the biological replicates for the insecticides are at different concentrations. ie each pool of 30 individuals was treated with one of three concentrations of the given insecticide, so there is only one biological replicate of Imidacloprid at 0.1 mg/L, one biol replicate of Imidacloprid at 0.01 mg/L, and one biol replicate of Imidacloprid at 0.001 mg/L etc. If this is true, there are not 3 biological replicates, but only one. I’m similarly unclear about the other treatments and life stages.
Section 4.3 Please include information about where in the gene the primers were designed, it would be great if the primers were in different exons or if one primer annealed across the junction of two exons to ensure no gDNA is amplified. Please state this clearly.
Section 4.4: I don’t follow- what are the eight experimental samples?
Section 4.5: This section does not provide enough detail about the transcriptome dataset that the CYP6a1 FPKM values for nymphs and adults were drawn from. Table S1 doesn’t refer to CYP6a1 – neither the contig, ORF position or size, or the primer sequences.
Comments on the Quality of English LanguageThere were a number of grammatical mistakes throughout (I have pointed out some in my comments above).
Reviewer 2 Report
Comments and Suggestions for Authors
Dear authors,
Your manuscript is certainly interesting from the point of view of the plant as support material, the chosen pest species and you have convinced me of the importance in the molecular mechanism of insecticide resistance formation.
However, I have doubts about the impact and practical applicability of the results because you have tested insecticides that are no longer used in most parts of the world, such as imidacloprid and thiamethoxam from the neonicotinoid group, which have been shown to be harmful to humans in 2018. In fact, it is a sensitive issue worldwide, at least at the European and American level.
In my opinion, the results should be applicable to a wider area, if possible on a global scale. But if these chemicals are still allowed in Asia (which I can't comment on because it's a different geographical area) and you are happy with the results, I have nothing against other reviewers coming to a positive decision.
In addition to this (important) aspect, other minor clarifications are needed.
Lines 53-55/Figure 1: Clarification on whether the images in Figure 1 are original. I can't tell if what you presented in the Introduction is original or from other sources (usually information from others is reproduced in this first chapter). If they are from other researchers, then you should attribute them.
Lines 392-397: The part of the description of the basic material (the pest) does not include enough information about the quantity (number of adults and nymphs reared in the laboratory), the period of their harvest from the field, the culture conditions (organic or treated) of the surfaces where the individuals were collected. All this, as answers added to the manuscript, will complete the methodology and make it more credible to be carried out.
More specifically, you should make the connection between the methodological steps/activities, i.e. there is no clear line separating individuals harvested from the field from those harvested/subjected to experiments on plants grown in the laboratory. Place the collection activity in time (when did you do this, in what year, period?)
Lines 381-389: Your conclusion is that... These results are useful to further elucidate the molecular mechanism of insecticide resistance in B. gobica.... Are you sure that you can generalise to all insecticides of all classes or are you referring strictly to imidacloprid?
Perhaps it would be good to make a conclusive reference to the other insecticides you have also studied, or take each one separately. Something is missing in your statements from the conclusions to the results.
Kind regards,
R
